# Targeting Toxins toward Tumors

**DOI:** 10.3390/molecules26051292

**Published:** 2021-02-27

**Authors:** Henrik Franzyk, Søren Brøgger Christensen

**Affiliations:** Department of Drug Design and Pharmacology, University of Copenhagen, Universitetsparken 2, DK-2100 Copenhagen Ø, Denmark; henrik.franzyk@sund.ku.dk

**Keywords:** chemotherapy, prodrug, drug targeting, overexpressed enzymes, ADC, ADEPT, GDEPT, LEAPT, PROTAC

## Abstract

Many cancer diseases, e.g., prostate cancer and lung cancer, develop very slowly. Common chemotherapeutics like vincristine, vinblastine and taxol target cancer cells in their proliferating states. In slowly developing cancer diseases only a minor part of the malignant cells will be in a proliferative state, and consequently these drugs will exert a concomitant damage on rapidly proliferating benign tissue as well. A number of toxins possess an ability to kill cells in all states independently of whether they are benign or malignant. Such toxins can only be used as chemotherapeutics if they can be targeted selectively against the tumors. Examples of such toxins are mertansine, calicheamicins and thapsigargins, which all kill cells at low micromolar or nanomolar concentrations. Advanced prodrug concepts enabling targeting of these toxins to cancer tissue comprise antibody-directed enzyme prodrug therapy (ADEPT), gene-directed enzyme prodrug therapy (GDEPT), lectin-directed enzyme-activated prodrug therapy (LEAPT), and antibody-drug conjugated therapy (ADC), which will be discussed in the present review. The review also includes recent examples of protease-targeting chimera (PROTAC) for knockdown of receptors essential for development of tumors. In addition, targeting of toxins relying on tumor-overexpressed enzymes with unique substrate specificity will be mentioned.

## 1. Introduction

According to the International Union of Pure and Applied Chemistry (IUPAC), prodrugs are defined as chemically modified drugs that undergo biological and/or chemical transformation(s) before eliciting pharmacological responses [1]. Drugs may be converted into prodrugs in order to: (i) increase their bioavailability, (ii) target the drugs toward tissues such as tumors, (iii) decrease toxicity, (iv) increase chemical stability, (v) increase solubility, or (vi) mask unpleasant taste [2,3]. Prodrugs are formed by covalent attachment of the drug to a carrier, also often termed as the promoiety, which is subjected to cleavage within the body to release the active drugs. Promoieties and their degradation products should be nontoxic and nonimmunogenic [2]. Pharmacologically inactive compounds, which in the organism are modified into active drugs, are known as bio-precursor prodrugs. Examples of such prodrugs are proguanil that in the liver is converted into the antimalarial drug cycloguanil [4], salicin that is converted into salicylic acid [5], and acetanilide that is converted into acetaminophen [3]. Both salicylic acid and acetaminophen are antipyretics and analgesics. Another class of prodrugs is the co-drugs that consist of two drugs covalently attached to each other—either directly or via a linker in a way so that they act as promoieties for each other [2,6]. Examples of co-drugs comprise sulfasalazine, which in the body is degraded to 5-aminosalicylic acid and sulfapyridine [6,7], and benorylate, which is an ester of acetylsalicylic acid with paracetamol [6]. 

A review focused on the prodrug approach revealed the state of the art in 2017 [8]. In the present review new developments in the fields of boronic acids as prodrugs, of anthracyclines, and of antibody–drug conjugates, targeting of paclitaxel and refined use of prostate-specific membrane antigen (PSMA) for delivery of payloads are described. The problem of targeting oncogenes for neoplastic tissue in gene-directed enzyme prodrug therapy is discussed (which often has been omitted in previous reviews). Finally, a new group of prodrugs, which link two pharmacophores, i.e., PROtease Targeting Chimeras (PROTACs; [9,10]) and lectin-directed enxym activated prodrugs, are discussed. One of these pharmacophores has a high affinity for an E3 ubiquitin ligase (an enzyme that, with assistance of an E2 ubiquitin-conjugating enzyme, transfers ubiquitin to its protein substrate), while the other has an affinity for the targeted protein e.g., a receptor or ion channel. After binding of the protein and the ubiquitin ligase, the biomolecule will be modified with ubiquitin and subsequently cleaved by a 26S proteasome, which degrades ubiquitinated proteins [9,10]. 

Thus, the present review comprises the following: (i) prodrugs cleaved in the acidic microenvironment of cancer cells (Section 2.1.1), (ii) prodrugs cleaved by reactive oxygen species (ROS) in cancer cells (Section 2.1.2), (iii) prodrugs cleaved by glutathione (Section 2.1.3), (iv) prodrugs cleaved by enzymes overexpressed in cancer cells ( Section 2.1.4), (v) prodrugs cleaved by glucuronidase (Section 2.1.5), (vi) prodrugs cleaved by prostate-specific antigen (PSA) or PSMA (Section 2.1.6), (vii) antibody–drug conjugates (Section 2.2), (viii) antibody-directed enzyme prodrug therapy (Section 2.3), (ix) gene-directed enzyme prodrug therapy (Section 2.4), (x) lectin-directed enzyme-activated prodrug therapy (Section 2.5), and (xi) protease-targeting chimeras (Section 3).

## 2. Prodrugs

### 2.1. Targeting by Selective Cleavage of Prodrugs in the Microenvironment of Cancer Cells

The metabolism in cancer cells involves a high rate of anaerobic glycolysis resulting in an overproduction of lactic acid and carbonic acid. Since the acid protons of these acids are exported into the extracellular medium, intracellular pH of cancer cells typically is higher (pH 7.4 versus 7.2 in normal cells). [8,11,12]. As a consequence of the proton transport the microenvironment surrounding cancer cells has a pH of 6.8 in contrast to the is estimated pH of blood 7.4 [13]. A high rate of glycolysis without oxygen supply causes hypoxia inside cancer cells [8,11,14,15]. The ensuing rise in the intracellular level of reactive oxygen species (ROS), such as hydrogen peroxide, formed during hypoxia conditions may be employed as a mode of targeting drugs toward cancer cells. Cancer cells also have an increased level of glutathione [16], β-glucuronidase [17], and some specific proteolytic enzymes [17,18,19].

#### 2.1.1. Prodrugs Cleaved in Acidic Media 

Salts of dithiocarbamates (e.g., **1**) are stable at physiological pH, but they will after protonation in the acidic microenvironment surrounding cells undergo cleavage to release the free amine (**2**) and carbondisulfide (Scheme 1). In a similar way the emitine monoamide (**3**) with 2-methylmaleic acid was used to simultaneously increase the solubility of emitine in water and enable its release from the prodrug in slightly acidic media [20].

A prodrug of doxorubicin (DOXO-EMCH, **4**) is linked via a hydrazone promoiety to a maleimide *N*-substituted with a 6-aminohexanoic spacer. This prodrug is designed to enable reaction with Cys-34 of serum albumin (to give **5**), present as the most abundant protein in blood. The resulting albumin-linked drug (**5**) cannot penetrate into cells until the hydrazone is hydrolyzed in the acidic environment around cancer cells to provide the free doxorubicin (**6**; Scheme 2). The clinical phase III trial, however, did not enable registration of the compound as a drug [14,21,22]. Despite these preliminary results, attempts to overcome the cardiotoxicity of DOX-EMCH continue.

Polyethylene glycol-coated liposomes encapsulating doxoxrubicin (Caelyx) show less cardiotoxicity than doxorubicin itself. The drug is used for treatment of breast cancer and ovarian cancer [23].

Paclitaxel is widely used for treatment of various cancers; however, poor solubility limits its use. The formulation vehicle Cremophor EL (CrEL; macrogolglycerol ricinoleate; polyethylene glycol (PEG)-35 castor oil) and ethanol are used to increase its solubility, but unfortunately CrEL causes side effects like hypersensitivity, neurotoxicity and nephrotoxicity [24]. These side effects have been overcome by a new formulation of lyophilized paclitaxel with serum albumin (i.e., Abraxane^®^) that provides nanoparticles (average size: 130 nm). This drug formulation has been approved by the U. S. Food and Drug Administration (FDA) for treatment of pancreatic cancer and non-small cell lung cancer (NSCLC) [25]. Moreover, paclitaxel (**7**) has been attached to nanoparticles via acetal linkages (**8**). Acetals are cleaved in the acidic environment of cancer cells (Scheme 3) [26]. This method is also used for conjugation of paclitaxel to nanoparticles prepared from polyethylene glycols (PEGs) [13].

#### 2.1.2. Prodrugs Cleaved by ROS

In normal cells, ATP is primarily produced by oxidative phosphorylation, whereas in cancer cells ATP primarily is produced by anaerobic glycolysis (the Warburg effect) [27,28]. The anaerobic pathway stimulates generation of ROS such as hydrogen peroxide [28]. The presence of hydrogen peroxide under certain physiological conditions can be used to facilitate cleavage of arylboronic acids, or esters thereof, to give phenols and boric acid [15]. The concentration of hydrogen peroxide in benign cells is estimated to approx. 1 µM, but in cancer cells it may reach even 10 µM [15]. Some boronic acids are oxidized by cytochrome P450 [15]. Boronic acids may also be oxidized by peroxynitrite [15]. It may be questioned whether arylboronic functionalities act as true promoieties, since the oxidation provides boric acid and the corresponding phenol. As an example, camptothecin-10-boronic (**9**) acid is oxidatively cleaved by hydrogen peroxide to give 10-hydroxycamptothecin (**10**; Scheme 4) [29]. The resulting 10-hydroxycamptothecin (**10**) proved to be a more potent topoisomerase inhibitor, and to be more cytotoxic in a number of cell systems than the original drug. Furthermore, this hydroxylated derivative exhibited tumor growth inhibition in xenograft models [29]. 

Arylboronic acid prodrugs of doxorubicin (e.g., **11**) have also been reported. Oxidative cleavage of the boronic acid moiety releases a phenol that spontaneously cleaves itself from the self-immolative spacer 4-hydroxybenzyl carbamate (**12**; Scheme 5) [15]. Doxorubicin (**6**) very efficiently kills cancer cells, however, a severe cardiotoxicity limits its use as a chemotherapeutic drug [30]. Targeting of the drug may reduce this side effect. The prodrug was found to induce regression of pancreatic tumors in mice, and further analysis revealed that the prodrug was cleaved to doxorubicin inside tumors (Scheme 5) [31]. 

Similarly, an aryl boronic acid prodrug (**13**) of paclitaxel (**7**), also containing a self-immolative linker, has been reported (Scheme 6) [15]. The size of the PEG moiety was adjusted so that the prodrug self-assembles into micelles with a size of ca. 50 nm. Native paclitaxel (**7**) was only released in the acidic microenvironment of the cells containing a high level of ROS. Consequently, reduced toxicity of the prodrug as compared to treatment with paclitaxel was observed in mice while retaining similar tumor regression [32]. At present no boronic acid prodrugs have been approved, despite intensive research being performed in the field [15].

#### 2.1.3. Prodrugs Cleaved by Glutathione

Glutathione (H-γGlu-Cys-Gly-OH) is present in almost all mammalian tissues, but usually it is overexpressed in cancer cells [33]. The active functionality of glutathione is the thiol, which enables the molecule to participate in redox reactions, and may thus protect the cell from a high level of ROS [33]. The molecule is able to cleave disulfides including linkages within prodrugs. Hence, this feature has been utilized in the construction of prodrugs attached to a promoiety via a disulfide linkage, and e.g., camptothecin (**14**) has been linked to a near-infrared (NIR) dicyanomethylenebenzopyran fluorophore (**15**; Scheme 7). An in vivo experiment using a mouse BCap-37 tumor xenograft model showed a significantly improved regression of tumors on mice treated with the prodrug as compared to that found for those treated with camptothecin or another prodrug in which the linkage consisted of a stable carbon–carbon bond instead of the disulfide bond [16].

#### 2.1.4. Prodrugs Cleaved by Expressed Enzymes

A number of enzymes are overexpressed in cancer cells. These enzymes include oxidoreductases, hydrolases, and matrix metalloproteinases (MMPs) [8]. A prodrug based on the overexpression of MMPs was designed for doxorubicin (Scheme 8) [34]. The peptide promoiety conjugated to the amine in doxorubicin prevents entry into cells, and consequently the compound is harmless to benign cells. By contrast, in the microenvironment of cancer cells the peptide is cleaved at the Gly-hPhe position. Proteases subsequently remove the remaining amino acids. In a HT1080 xenograft mouse preclinical model, the prodrug was more efficient in reducing tumor growth than doxorubicin itself, and less undesired toxicity was observed [34].

Cathepsin B is involved in cancer invasion and metastasis, and it is overexpressed in cancer tissue [35]. A prodrug (**17**) consisting of doxorubicin conjugated via the self-immolative linker 4-aminobenzyl alcohol to a dipeptide fragment (Ac-Phe-Lys-OH), which is a substrate for cathepsin B, has been designed (Scheme 9). In a mouse model this prodrug inhibited development of peritoneal carcinomatosis as well as its progression more efficiently and with fewer side effects than doxorubicin itself [35].

#### 2.1.5. Prodrugs Cleaved by β-glucuronidase

Glucuronides are formed as a phase II metabolism of drugs. β-Glucuronidases are expressed excessively in a number of tumors such as breast, lung and gastrointestinal tract carcinomas as well as in melanomas, where they are particularly abundant in necrotic areas [17,36,37]. Prodrugs based on this selective enzyme distribution include glucuronides of doxorubicin (**6**) and 4′-epi-doxorubicin (**19**) [17]. A self-immolative linker was introduced, since the β-glucuronide (i.e., **18**) conjugated directly to the sugar part of doxorubicin was not a substrate for β-glucuronidase (Scheme 10) [17].

In an attempt to circumvent the poor solubility of paclitaxel (**7**) in water a glucuronide (i.e., **21**) was made. The resulting glucuronide (**21**) proved indeed to be soluble in water, and it was rapidly converted into paclitaxel (**7**) in the presence of high concentrations of β-glucuronidase (Scheme 11) [38]. 

Likewise, 7-Aminocamptothecin was conjugated to glucuronic acid via another self-immolative linker (Scheme 12).

#### 2.1.6. Prodrugs Cleaved by PSA or PSMA

Prostate cancer is a slowly developing cancer disease. In high-income countries it is the cancer disease that causes second-most deaths among men. In its initial stages the disease can be treated with androgen ablation therapy. However, if progression of disease occurs during treatment with anti-androgens, resulting in development of distant metastasis, the prostate cancer is defined as metastatic Castration-Resistant Prostate Cancer (mCRPC) which is not sensitive to hormone ablation [40]. All tumors of mCRPCs secrete the enzyme prostate-specific antigen (PSA) into their microenvironment. PSA is a chymotrypsin-like protease with a unique substrate specificity [18]. PSA also diffuses into the bloodstream, but PSA in the blood is inactivated by complexing with blood proteins like serum albumin [40]. Conjugation of cytotoxins with different selectively labile peptides has been used for targeting of mCRPCs. Thus, *O*-desacetylvinblastine (**24**) has been targeted to mCRPCs by conjugation to a PSA-specific peptide substrate to give a prodrug (i.e., **25**) (Scheme 13) [19]. 

PSA cleaves the peptide between the two Ser residues adjacent to the C-terminal Pro residue, whereupon a spontaneous intramolecular attack of the amino group of the terminal Ser on the Pro carboxylate affords desacetylvinblastine (**24**) and a diketopiperazine. This intramolecular diketopiperazine formation appeared to depend on the presence of the Pro residue, as it did not occur when Leu was incorporated instead. Desacetylvinblastine (**24**) proved equally efficient in inducing mitotic arrest as the original vinblastine (Scheme 13) [19].

A number of peptides have been conjugated to doxorubicin (**6**) [41]. Among these different promoities the peptide H_2_N-Glu-Hyp-Ala-Ser-Chg-Ser-Leu-OH was found to afford a prodrug that was efficiently cleaved by PSA, and it showed a dramatically increased activity in reducing LNCaP xenografts in mice as compared to that of native doxorubicin (**6**). The released active drug consists of a mixture of doxorubicin and H-Leu-doxorubicin (Scheme 14) [41].

A drawback of using vinblastine or doxorubicin (**6**) as drugs for treatment of prostate cancer is that both compounds cause mitotic arrest, and consequently they primarily target proliferating cells. A more pronounced cell death is expected when toxins capable of killing cells in all stages are used. Thapsigargin (**27**) is a cytotoxic compound that kills cells in all states by blocking the sarco/endoplasmic reticulum Ca^2+^ ATPase (SERCA), thereby inducing the unfolded protein response leading to apoptosis [42]. The general toxicity of the compound requires targeting via a prodrug in order to avoid general systemic toxicity [43]. No obvious anchoring point for conjugation to a peptide exists in native thapsigargin, but replacement of the butanoyl moiety with a 12-aminododecanoic acid spacer (to give **28**) allows for introduction of an amine functionality that may serve as attachment point for a peptide promoiety (Scheme 15).

Conjugation of a thapsigargin analogue to a polar peptide was expected to inhibit diffusion into cells, thereby preventing the prodrug from reaching the intracellular SERCA pump. The hexapeptide H-His-Ser-Ser-Lys-Leu-Gln-OH is very efficiently released from the prodrug by PSA, but only to a limited extent by any other proteases in the human body [18]. The C-terminal Leu residue was introduced in order to make the prodrug a substrate for PSA. In vivo experiments in mice confirmed that the prodrug was only cleaved in the blood to a limited extent, but very efficiently cleaved by PSA to release the active Leu derivative within tumors. Thus, this prodrug prevented growth of tumors in mice [44]. 

KLK2 is another enzyme (previously named human kallikrein 2, hK2) that may be used for targeting drugs toward prostate cancers [45,46]. Similarly to PSA, KLK2 is also secreted from the prostate and prostate cancer cells. The level of KLK2 in blood can be used as a biomarker for prostate cancer, and KLK2 is inactivated upon entering the bloodstream by binding to blood proteins [47].

An alternative enzyme, characteristic for the prostate glandule, is prostate-specific membrane antigen (PSMA). The catalytic site of this enzyme extends outwards into the extracellular environment. The enzyme is not only expressed in the prostate glandule, but also in human prostatic carcinoma and in neovascular tissue of a number of tumors [48,49]. In a healthy individual the enzyme is exclusively expressed in the prostate, ensuring that cleavage of the peptide conjugated to its payload solely occurs in the prostata glandula [49,50,51,52,53]. As the only enzyme outside the central nervous system PSMA cleaves the amide linkage in the γ-Glu tetramer [54]. Taking advantage of this feature, a prodrug of the 12-aminododeanoate of desbutanoylthapsigargin (**28**), (i.e., mipsagargin, **31**) was made (Scheme 16). The C-terminal βAsp residue was introduced in order to make the prodrug a substrate for PSMA [49]. This prodrug, which has been prepared by solid-phase synthesis [55], was cleaved rapidly in tumors to release the βAsp derivative, which slowly was cleaved to provide the free mipsagargin. A solid-phase synthesis of these guaianolide prodrugs were developed [55].

In the clinical phase II trial this prodrug conferred a prolonged stabilization of the disease in patients with hepatocellular carcinoma [56]. Hepatocellular cancer also expresses PSMA in neovascular tissue, which thus should be sensitive to mipsagargin [57]. However, the results obtained in clinical phase II trial did not meet the expectations, and hence the drug was not marketed [58]. A major problem might be that the prodrug, despite its five negative charges on its peptide side chains, has been found to be able to penetrate cell membranes in benign cells, thereby causing unspecific toxicity [59]. 

### 2.2. Antibody–Drug Conjugates (ADCs)

An antibody–drug conjugate (ADC) is a prodrug consisting of a monoclonal antibody conjugated to a cytotoxin (i.e., the payload) via a linker. The prodrug is designed based on the hypothesis that appropriate antibodies that preferentially bind to cancer-specific antigens (located on the surface of cancer cells) can be obtained. Indeed, the two first ADCs, Adcetris^®^ (brentuximab vedotin) for Hodgkin lymphoma and Kadcyla^®^ (trastuzumab emtansine) for breast cancer, were approved by the FDA in 2011 and in 2013, respectively. Since then six additional ADCs have been approved by the FDA (i.e., Besponsa^®^ (inotuzumab ozogamicin), Mylotarg^®^ (gemtuzumab ozogamicin), Polivy^®^, Enhertu^®^, Padcev^®^, and Trodelvy^®^) [61]. More than 60 ADC prodrugs are under clinical development [62,63,64].

The linker should be stable in circulation to avoid release of free cytotoxin causing systemic toxicity, as is often seen with conventional chemotherapeutics (Figure 1). 

Monoclonal antibodies of rodent origin may cause severe immunogenic reactions in humans. The use of chimeric humanized antibodies has to some extent solved this problem [65].

The ideal cytotoxin should be very toxic since only a limited number of antigens are present on the surface of malignant cells, and consequently only a limited number of ADCs can be internalized to exert the cytotoxic effect [66]. A number of toxins have been used as payloads, e.g., the peptide monomethyl auristatin E, the polyketide macrolides ansamitocin (as a mixture of aliphatic alkyl esters) and maytansins (= maitansins) as well as doxorubicin (**6**), duocarmycin (**32**), and enediynes like calicheamicin γ_1_^I^ (**33**) have been converted into ADCs (Figure 2) [61,63]. 

The calicheamycins is a group of extremely potent cytotoxins. Originally the compounds were tested as antibiotics, and their minimum inhibitory cancentration (MIC) values were found to be from 0.5 µg/mL toward *Eschericia coli* to less than 0.2 ng/mL against *Bacillus subtilis* [67]. Antitumor activity was tested against P388 leukemia and B16 melanoma in mice by intraperitoneal injection. The optimum dose was found to be 5 µg/kg as compared to 1.6 mg/kg for cisplatin against P388 cells, and 1.25 µg/kg as compared to 300 mg/kg for cisplatin against B16 cells [67]. All mice died, indicating a high general toxicity. Calicheamicin γ_1_^I^ is an interesting payload because of its extreme cytotoxicity [66]. This molecule represents an extraordinary example of natural bioengineering that has occurred during evolution. The sugar part including the iodinated aromatic residue confers affinity for DNA. After complexing with DNA, a nucleophilic attack, e.g., by glutathione on the central sulfur in the trisulfide, leads to the cleavage of this linkage, thereby releasing the free thiol, which undergoes an intramolecular thiol Michael addition to the α,β-unsaturated ketone. By changing the trigonal bridgehead β-carbon to a tetragonal carbon, sufficient tension is induced in the 10-membered ring to initiate a Bergman cyclization. The resulting intermediate diradical finally cleaves the DNA [66] (Scheme 17). 

In particular two functional groups are used for conjugation of drugs via a linker to the antibody, namely the thiol group of cysteine and/or the amino groups of lysine residues [61]. Traut’s reagent or carbodiimides with or without hydroxysuccinimide have been used for coupling of a carboxylic acid to lysine side chains [63]. As an example, Scheme 18 depicts how calicheamicin γ_1_^I^ (**33**) has been coupled to an antibody in Mylotarg (gemtuzumab ozogamicin) [63].

After internalization of the antibody–antigen complex, enzymes within the cell facilitate hydrolysis of the hydrazone moiety [63]. The disulfide will be cleaved by glutathione, enabling Bergman cyclization. Other linkers have been designed to be cleaved by intracellular proteases like cathepsin B (e.g., Adcetris). After internalization of the ADC Kadcyla, in which mertansine (**34**) is linked to an antibody, the linker including the lysine residue remains attached to the payload after decomposition of the antibody (Scheme 19). This extended linker moiety appears not to compromise the effect of the drug. Other examples of ADCs have been reported by Nicolaou and Rigol [63].

### 2.3. Antibody-Directed Enzyme Prodrug Therapy

Similarly to ADCs, the concept of antibody-directed prodrug therapy (ADEPT) is based on the ability of antibodies to selectively target antigens expressed abundantly on the surface of cancer cells [68,69]. The principle involves a preferential binding of a non-human enzyme to the surface of cancer cells via an antibody–antigen complex. The choice of a non-human enzyme makes it possible to choose a linker which solely is cleaved by this enzyme and not by any endogenous enzymes. On the other hand, a potential drawback of using a non-human enzyme may be a strong allergic reaction due to unforeseen immunogenicity. In contrast to the ADC approach, non-internalizing antigens can be targeted. The enzyme is linked to the antibody by using a bisfunctional linker, where one functionality can be linked to the lysine side chains present on the enzyme, while the other functionality can be linked to thiols of cysteines on the antibody (Figure 3) [68,69,70]. After administration to the patient the antibody binds to the surface of the cancer cell. When the excess free antibody–enzyme conjugate is completely cleared from the body the prodrug is administered.

As mentioned above, a prerequisite for the use of enzymes not present in the human body is that they are non-immunogenic [70]. Enzymes belonging to the families of alkaline phosphatases (cleaving phosphate from prodrugs), peptidases, sulfatases (for cleavage of sulfate monoesters), carboxylesterases and carboxypeptidases (for cleavage of e.g., glutamic amides), have been investigated in this respect [70].

Some antibodies themselves possess catalytic properties, e.g., the antibody 38C2 catalyzes retroaldol and retro-Mannich reactions; for example, a prodrug of doxoxrubicin is cleaved by 38C2 (Scheme 20). [70,71].

Even though some promising clinical results have been obtained, no drugs based on ADEPT are in clinical use at present [68,69].

### 2.4. Gene-Directed Enzyme Prodrug Therapy (GEPDT)

In gene-directed enzyme prodrug therapy (GEPDT), a gene encoding for a unique enzyme is introduced into the tumor cells by using a vector. The technique was already introduced in 1986, and the gene introduced into cells is called a suicide gene [72]. Upon expression of the enzyme on the cancer cell surface, the enzyme enables cleavage of the linkage between the payload and the promoiety, after which the payload may diffuse into the cancer cell (Figure 4) [69,70]. A major drawback in GDEPT is the prerequisite of achieving selective transfer of a gene into malign cells. Retroviruses, adenoviruses and herpes viruses have been studied as potential vectors [73]. Retroviral vectors have some selectivity, since they are only incorporated into the genome of actively dividing cells [73]. Attachment of tissue-specific promoters may allow for transgenic expression only in neoplastic cells. The use of receptor-specific vectors has also been proposed [73]. Mesenchymal stem cells, exhibiting strong tropism toward tumors and metastases expressing receptors on their surface, can efficiently be transduced with vectors [72]. Virus-like particles have also been used to internalize the gene into cells [74]. No drugs based on the principle of GDEPT have been approved so far.

### 2.5. Lectin-Directed Enzyme-Activated Prodrug Therapy (LEAPT)

In lectin-directed enzyme-activated prodrug therapy (LEAPT), a drug or an enzyme is targeted toward cancer cells by using sugar–protein recognition, whereas antigen–antibody recognition is used for targeting in ADEPT and GDEPT. Lectins are proteins involved in biological carbohydrate recognition comprising cellular processes such as growth, differentiation, proliferation or apoptosis [75]. In order to improve the selectivity of doxorubicin, a galacturonamide derivative (i.e., **36**) was prepared (Figure 5) [76]. Here, the expression of asialoglycoprotein receptors (ASGPRs) 1 and 2 on the surface of hepatocytes with a high affinity for D-galactose and L-rhamnose was exploited. After binding to the receptor, the appropriate carbohydrate-containing ligand is internalized. The ASGPRs are expressed on the surface of HT-29, MCF-7 and A549 cells to a much higher extent than in normal liver cells [76]. The Gal-Dox derivative proved to exhibit higher selectivity toward cancer cell lines than doxorubicin (**6**) itself. In S180 tumor-bearing mice, the Gal-Dox-treated group had a higher accumulation of the drug in the malignant tissue than the doxorubicin-treated group as well as an improved survival rate [76]. However, in this case the doxorubicin derivative may in fact not be a true prodrug, since probably the Gal derivative may also interact with the topoisomerase target.

Another approach utilizes the overexpression of glucose transporters (GLUT) in cancer tissue. By preparing glucose or glucuronic acid derivatives of paclitaxel, two advantages are obtained: (i) the compounds become more soluble in water, and (ii) increased uptake through the GLUT into the cancer tissue (Figure 6) [77].

The glucose and glucuronic acid derivatives (**37** and **38**) were found to exert a low cytotoxicity on benign cells, but an activity similar to that of paclitaxel (**7**) itself on cell lines expressing GLUT. It is assumed that the prodrug is cleaved by intracellular β-glucosidases. A mechanism involving cleavage of the methyl glucoside followed by self-immolative cleavage to give paclitaxel (**7**) has been proposed [77]. However, the glucose derivative reported was an α-glycoside.

A two-phase LEAPT mechanism overcoming the requirement for intracellular cleavage of the prodrug has been suggested. In the first phase a glycosylated enzyme interacts with a carbohydrate-recognizing lectin on the surface of the cells in the targeted tissue. After similar interactions, a glycosylated prodrug, which is a substrate for the glycosylated enzyme, becomes internalized as well. Inside the cells, the internalized enzyme cleaves the glycosylated prodrug to liberate the active drug (Scheme 21).

A procedure for pergalactosylation of a naringinase produced by *Penicillium decumbens* has been developed to give a pergalactosylated enzyme, which showed high affinity for ASGPRs on the surface of hepatocytes. Binding to ASPGR triggers internalization of the bound ligand. The naringinase possesses α-rhamnosidase and β-glucosidase activities. Thus, a rhamnose derivative of doxorubicin (**6**) was prepared, and the stability of this derivative was tested [78]. At present, no drugs based on the LEAPT principle have been approved.

## 3. Protease-Targeting Chimeras (PROTAC)

In living cells, misfolded, damaged or mutated proteins are removed from the cells by natural processes, in which the protein first becomes covalently bound to one of a number of ubiquitins, which are highly conserved 76-residue peptides [79]. This conjugation process involves ubiquitin-activating enzymes E1, which transfer ubiquitin to E2 from where ubiquitin is transferred to a von Hippel–Lindau(VHL)-cullin-RING ligase complex including E3 that conjugates ubiquitin to the target protein mainly via lysine residues [9,10]. Subsequently, the ubiquitin-modified protein is degraded by a 26S proteasome to give a number of small peptides and a number of lysine-modified ubiquitins [9,10,80]. Other proteases finally cleave the oligopeptides into free amino acids [81].

Advantages of this system are that drugs may be developed to selectively remove intracellular proteins. A chimera consisting of a residue with high affinity for the VHL complex was via a linker attached to a moiety with high affinity for the estrogen-related receptor α (ERRα). After complexing with VHL, knockdown of the ERRα level was observed. The first experiment was performed in MCF7 cells after incubation with the chimeras to knock down ERRα [9]. Moreover, a serine-threonine kinase (RIPK2) was knocked down after incubation of MCF7 cells with a chimera consisting of a moiety with high affinity for the VHL complex (Figure 7) [9].

The effect of small-molecule drugs as ligands for biomolecules in treatment of cancer diseases can be limited by mutations in the gene encoding the biomolecule, thereby making the modified target insensitive to the agent. Such mutations are observed for the epidermal growth factor receptors and androgen receptors [82]. PROTAC has been used to enable knockdown of steroid receptors and for non-small lung cancer by knockdown of epidermal growth factor. In addition, the anaphylactic lymphoma kinase can similarly be removed as a possible treatment of different types of human cancers [82]. The PROTAC technique is still at an early stage, and at present no such drugs are currently in clinical use.

## 4. Conclusions

In recent decades, several diverse methods have been developed for the targeting of toxins to cancer tissue to avoid their general systemic toxicity. In the present review, the initial sections concern new attempts developed for prodrugs to be cleaved predominantly in the microenvironment of cancer cells and tumors. Thus, the lower pH characterizing cancer tissue has been explored for selective cleavage of prodrugs based on amides of a substituted maleic acid [20], a hydrazone promoiety [14,21,22], and labile acetal linkage to polymers [26]. A prodrug of doxorubicin (Aldoxorubicin) designed to prevent cardiotoxicity expected to be cleaved by the acidic microenvironment of cancer cells failed in clinical trial III [22]. Moreover, proof-of-concept studies of prodrugs relying on selective cleavage due to the increased ROS production in cancer cells comprise examples of arylboronic acid derivatives [13,15,29]. An example of glutathione-promoted cleavage of a disulfide-based prodrug has also appeared [16].

Next, enzymes, overexpressed by cancer cells or neovascular tissue in tumors, capable of selective cleavage of prodrugs carrying a peptide substrate moiety, offer several examples: e.g., MMP [34], cathepsin B [35], β-glucuronidase [17,38,39], and PSA [19,41,43] and PSMA [56,60]. One prodrug, mipsagargin, actually went into clinical phase III trials but, despite the polarity of the γGlu-γGlu-γGlu-γGlu-βAsp peptide moiety, the compound appeared to be able to penetrate cell membranes of benign cells also, and thus cause general toxicity [58].

In addition, the progress within the field of ADCs (with an anticancer drug as payload) comprise >60 entities in clinical development [62,63,64]. In total eight ADCs have been approved by the FDA as new and improved cancer therapies, albeit not in the period 2014 to 2019 [83,84,85,86,87,88]. Calicheamicin and maytansine have been used as the payload in many of these drugs [63].

Another approach also involving antibodies is ADEPT [70,71]; however, even though promising clinical results have been reported, no drugs based on ADEPT are currently approved for clinical use [68,69].

In addition, an advanced approach requiring selective introduction of a gene, coding for an enzyme capable of cleaving a prodrug, into cancer cells (i.e., GDEPT) [69,70] is considered a promising approach, but so far no drugs based on this concept have been approved. Similarly, targeting to cancer cells via sugar–protein recognition processes involving lectins present on the surface of cancer cells (i.e., LEAPT) have been explored [76,77,78]. Nevertheless, no drugs based on these principles have been approved as yet.

Finally, recent examples of protease-targeting chimeras (PROTACs) involve ubiquitination enzyme complexes that undergo proteolytic degradation to release the drug [9,10,80], however, this technique is at an early stage, and no examples of its clinical use have appeared.

Even though the described techniques have been utilized to improve the solubility of paclitaxel and selectivity of doxorubicin, the associated prodrugs have not as yet shown sufficiently improved properties to convince medical agencies that they can be approved as drugs. In conclusion, the new approaches reviewed here may indeed lead to future new anticancer drugs that are urgently needed for treatment of cancer diseases for which no cure exists. Nevertheless, most of these recently developed targeting principles remain to result in approved drugs, which emphasizes the need for further research to unleash the full potential of these concepts currently considered for experimental therapies.

## Data Availability

No supporting information for this work.

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
