# Peer review of "Targeting Toxins toward Tumors"

_molecules, 2021, doi:10.3390/molecules26051292_

Round 1

Reviewer 1 Report

Recommendation: Publish after major revisions noted.

Comments:

This review manuscript by Christensen summarized the strategies developed for targeting toxins toward tumors. Activation of prodrugs under tumor-characteristic features, antibody-drug conjugates, ADEPT, GDEPT, LEAPT, and PROTAC were included. Representative prodrugs/drug-biomolecule complexes were employed to explain each strategy. However, the following items should be addressed prior to publication:

  1. The topic on targeting toxins toward tumors was quite well reviewed at the reference 10 published in 2017. Thus, the authors should emphasize ref 10 at the Introduction and clarify what new techniques have been developed after 2017 and what were not included in ref 10.

  1. A Contents would be welcomed by the audiences. However, if Contents is not allowed by Molecules, the authors should have a short summary, at the end of the Introduction, on what strategies will be discussed in this review.

  1. At first paragraph at the 2.1 section, the author should briefly describe the characteristic features (lower pH, ROS, glutathione…) existing at the cancerous cells/microenvironments, which will be employed for the cleavage of prodrugs selectively at cancerous cells/microenvironment. From line 55-62, only low pH and ROS were described.

  1. At line 57 replace the review reference [10] with original research publications.

  1. At each subsection under 2.1, the references on the cancer-characteristic features, how they are formed, and how they can be employed should be described.

  1. Number/Name the compounds at the Figures/Schemes to help the audience to locate them more easily.

  1. Name the reaction procedure as Scheme instead of Figure. e.g. Figure 1 should be Scheme 1.

  1. Combine sentence at Line 68-69 to paragraph at line 64-65. You don't separate two sentences into two independent paragraphs.

  1. References are needed for the statement at line 130 and line 157.

  1. Combine paragraphs at Line 194-196 and Line 202-207.

  1. When discussing the cytotoxicity of toxins, it would be helpful to include the IC50 of toxins on benign and cancerous cells from references. e.g. at line 284, calicheamicin gamma was stated to be of extreme cytotoxicity. A reported IC50 will help the audience to really learn how toxic it could be.

  1. At Figure 21, label the extracellular and intracellular sites; the bean-shaped structures used as receptors for antibody should be labeled.

  1. At Figure 22, label the extracellular and intracellular sites; the bean-shaped structures used as receptors for antibody should be labeled; the scissors should be label as enzyme; Figure 22 A is confusion: it seems like the prodrug enters the cell. Thus, show the activation of prodrug first outside cells and then the entering of active drug into the cells (Figure 22 B)

  1. At 2.4 GEPDT, how vector selectively invades cancerous cells/tissue is essential to direct the drug to tumor tissue. Nevertheless, related references on this selective invasion to cancerous cells were NOT clarified in this review. Please include.

  1. References are needed for line 359 and 360.

  1. As stated at Line 358, ASGPRs were also highly expressed at hepatocyte, which makes the ASGPRs-based LEAPT less persuasive. Wouldn't ASGPRs-based LEAPT lead to liver failure? Please cite related references and explain.

  1. References for Line 369.

  1. Reference for Line 385.

  1. Figure 28 was not well designed to elucidate PROTAC. Without the description at the text, the figure was not informative enough. Figures perfectly describing PROTAC are easily available from references and from Google. Suggestion: the author can just copy from somewhere and cite the source.

  1. In paragraph at line 408: is ERRa over-expressed at cancerous cells? If no, how PROTAC work selectively toward cancerous tissue? What is the effect of knockdown of the ERRa and RIPK2 on the variation of MCF7 cells? Please provide all related references.

  1. A more comprehensive conclusion is required. The conclusion needs to have a short summary of all the reviewed techniques applied to direct toxins to cancer tissues. Based on the authors' expertise on this field, the authors can provide their own opinions on the further developments of each techniques they reviewed in this review.

  1. Reference 7. No pp. given? Isn't it e516?

Author Response

Reviewer 1

Comments:

This review manuscript by Christensen summarized the strategies developed for targeting toxins toward tumors. Activation of prodrugs under tumor-characteristic features, antibody-drug conjugates, ADEPT, GDEPT, LEAPT, and PROTAC were included. Representative prodrugs/drug-biomolecule complexes were employed to explain each strategy. However, the following items should be addressed prior to publication:

  1. The topic on targeting toxins toward tumors was quite well reviewed at the reference 10 published in 2017. Thus, the authors should emphasize ref 10 at the Introduction and clarify what new techniques have been developed after 2017 and what were not included in ref 10.

Response: We agree with the reviewer that the specific subject of PROTAC was reviewed in depth in ref. 10 from 2017 (Zhang et al.; this review comprises 6 pages excluding references). However, this subject is only treated briefly in the current review [i.e., Section 3. Protease-targeting chimeras (PROTAC) constituting 1 page corresponding to pages 21-22 in the revised manuscript]. Moreover, we have focused on the most recent references (please notice that a third of all included references are from 2017 till now). Finally, we have inserted a sentence explaining the techniques not mentioned in the review of Zhang et al. 2017, but included in the present review. These changes have resulted in a altered numbering of reference so that  Zhang et al. 2017 now is ref. 8.

  1. A Contents would be welcomed by the audiences. However, if Contents is not allowed by Molecules, the authors should have a short summary, at the end of the Introduction, on what strategies will be discussed in this review.

Response: We have expenaded the Introduction section with a brief summary of techniques reviewed (referring to the associated paragraphs). If recommended by the Editor a more traditional Table of Contents may instead be inserted after the Keywords.

  1. At first paragraph at the 2.1 section, the author should briefly describe the characteristic features (lower pH, ROS, glutathione…) existing at the cancerous cells/microenvironments, which will be employed for the cleavage of prodrugs selectively at cancerous cells/microenvironment. From line 55-62, only low pH and ROS were described.

Response: The different characteristics for cancer cells have included in been paragraph 2.1.

  1. At line 57 replace the review reference [10] with original research publications.

Response: References to original publications describing the characteristics of tumor cells have been given.

  1. At each subsection under 2.1, the references on the cancer-characteristic features, how they are formed, and how they can be employed should be described.

Response: The characteristic features of cancer cells are described, and as far as possible a rationale as to how these arise are briefly presented.

  1. Number/Name the compounds at the Figures/Schemes to help the audience to locate them more easily.

Response: The suggested numbering (and proper reference to these in the text) has been implemented.

  1. Name the reaction procedure as Scheme instead of Figure. e.g. Figure 1 should be Scheme 1.

Response: Figures have been replaced with Schemes where relevant.

  1. Combine sentence at Line 68-69 to paragraph at line 64-65. You don't separate two sentences into two independent paragraphs.

Response: The sentence is now placed above the figure.

  1. References are needed for the statement at line 130 and line 157.

Response: References have been added.

  1. Combine paragraphs at Line 194-196 and Line 202-207.

Response: The sentence has been moved as suggested.

  1. When discussing the cytotoxicity of toxins, it would be helpful to include the IC50 of toxins on benign and cancerous cells from references. e.g. at line 284, calicheamicin gamma was stated to be of extreme cytotoxicity. A reported IC50 will help the audience to really learn how toxic it could be.

Response: Stating the IC50 value of calichemycin is not as simple as it might seem, since it is depending on other factors like the amount of thiols present in the media. Values and data from animal experiments have been added.

  1. At Figure 21, label the extracellular and intracellular sites; the bean-shaped structures used as receptors for antibody should be labeled.

Response: The requested labels have been added.

  1. At Figure 22, label the extracellular and intracellular sites; the bean-shaped structures used as receptors for antibody should be labeled; the scissors should be label as enzyme; Figure 22 A is confusion: it seems like the prodrug enters the cell. Thus, show the activation of prodrug first outside cells and then the entering of active drug into the cells (Figure 22 B).

Response: The requested labels have been added.

  1. At 2.4 GEPDT, how vector selectively invades cancerous cells/tissue is essential to direct the drug to tumor tissue. Nevertheless, related references on this selective invasion to cancerous cells were NOT clarified in this review. Please include.

Response: At present GDEPT is performed on cell cultures to show the validity of the principle. Selectivity toward different cell types has not been studied yet.

  1. References are needed for line 359 and 360.

Response: The references have been added.

  1. As stated at Line 358, ASGPRs were also highly expressed at hepatocyte, which makes the ASGPRs-based LEAPT less persuasive. Wouldn't ASGPRs-based LEAPT lead to liver failure? Please cite related references and explain.

Response: ASPGRs are expressed to a much higher extend on cancerous cells, and a reference is given.

  1. References for Line 369.

Response: References have been added.

  1. Reference for Line 385.

Response: Reference has been added.

  1. Figure 28 was not well designed to elucidate PROTAC. Without the description at the text, the figure was not informative enough. Figures perfectly describing PROTAC are easily available from references and from Google. Suggestion: the author can just copy from somewhere and cite the source.

Response: A new figure has been made.

  1. In paragraph at line 408: is ERRa over-expressed at cancerous cells? If no, how PROTAC work selectively toward cancerous tissue? What is the effect of knockdown of the ERRa and RIPK2 on the variation of MCF7 cells? Please provide all related references.

Response: The technique appears still to be in an early phase, and so far only cell cultures have been investigated.

  1. A more comprehensive conclusion is required. The conclusion needs to have a short summary of all the reviewed techniques applied to direct toxins to cancer tissues. Based on the authors' expertise on this field, the authors can provide their own opinions on the further developments of each techniques they reviewed in this review.

Response: An extended and revised conclusion has been made. This points to the fact that only a few of the prodrug approaches discussed have resulted in clinically used drugs, thus emphasizing the need for further research to unleash the full potential of the concepts currently considered for experimental therapies involving selective targeting of anticancer drugs.

  1. Reference 7. No pp. given? Isn't it e516?

Response: The missing article number has been added.

We appreciate the comments of the reviewer which have improved the clarity and the quality of the manuscript.

Reviewer 2 Report

Literature suggestions:

  • line 85: Abraxane is approved by FDA also for treatment of pancreatic cancer and NSCLC
  • line 88: PEG, associated with a liposomal formulation, is also used for the creation of a derivative of doxorubicin, namely pegylated liposomal doxorubicin (Caelyx). This drug is used for various types of cancer, such as breast cancer and ovarian cancer. I suggest to elaborate on the subject.
  • lines 186 and 187: the statement is quite uncorrect. The expression: "However, if progression disease (PD) occurs during treatment with anti-androgens, with the development of distant metastases, prostate cancer is definited as mCRPCs" seems more appropriate.
  • line 253:the link between PSMA and hepatocellular carcinoma should be better explained
  • line 264: you could add: "...by FDA the first, in 2011 for Hodgkin lynphoma, and the second in 2013 for breast cancer"

Editing suggestions:

  • line 119: pay attention to how "paclitaxel" is written: "paclitazel"
  • line 183: the name is PSMA not PMSA

Figures should be placed after the reference paragraph and not before. Such as the figures: 5, 8, 10 and 27.

Author Response

Reviewer 2

Comments and Suggestions for Authors:

Literature suggestions:

line 85: Abraxane is approved by FDA also for treatment of pancreatic cancer and NSCLC

Response: This fact has been added

line 88: PEG, associated with a liposomal formulation, is also used for the creation of a derivative of doxorubicin, namely pegylated liposomal doxorubicin (Caelyx). This drug is used for various types of cancer, such as breast cancer and ovarian cancer. I suggest to elaborate on the subject.

Response: This fact is now mentioned.

lines 186 and 187: the statement is quite uncorrect. The expression: "However, if progression disease (PD) occurs during treatment with anti-androgens, with the development of distant metastases, prostate cancer is definited as mCRPCs" seems more appropriate.

Response: The sentence has been rephrased as suggest.

line 253:the link between PSMA and hepatocellular carcinoma should be better explained.

Response: A reference has been added to explain that HCC express PSMA in neovascular tissue.

line 264: you could add: "...by FDA the first, in 2011 for Hodgkin lynphoma, and the second in 2013 for breast cancer".

Response: The fact has been mentioned as suggested.

Editing suggestions:

line 119: pay attention to how "paclitaxel" is written: "paclitazel"

line 183: the name is PSMA not PMSA

Figures should be placed after the reference paragraph and not before. Such as the figures: 5, 8, 10 and 27.

Response: Misspellings have been corrected and figures moved as suggested.

We appreciate the Reviewers comments, which have improved the clarity and statements in the mansuscript.

Round 2

Reviewer 1 Report

Recommendation: Publish after minor revisions noted.

Comments:

The manuscript was revised according to the reviewers' comments. It is better now. However, the following items should be addressed prior to publication:

1. Ref 8 (Ref 10 in the first version) didn’t discuss Protac. Ref 8 was mainly focused on the other strategies for cancer cell-specific targeting. The author should read the review and rewrite the sentences related.

2. The second paragraph at Conclusion is good. But the first paragraph is more like information for Introduction rather than for Conclusion. The conclusion needs to have brief summary of all the reviewed techniques applied to direct toxins to cancer tissues

Author Response

Reviewer 2 – comments (Round 2)

The manuscript was revised according to the reviewers' comments. It is better now. However, the following items should be addressed prior to publication:

  1. Ref 8 (Ref 10 in the first version) didn’t discuss Protac. Ref 8 was mainly focused on the other strategies for cancer cell-specific targeting. The author should read the review and rewrite the sentences related.

Response: This unfortunate mistake has been corrected so that the sentence now conveys that it is the general state of the art of the prodrug approach for anticancer drugs (rather than PROTACs) that was reviewed in ref. [8] in 2017.

Also, more clear references to PROTACS have now been inserted (i.e., [9,10]) in the appropriate sentence in the same paragraph.

  1. The second paragraph at Conclusion is good. But the first paragraph is more like information for Introduction rather than for Conclusion. The conclusion needs to have brief summary of all the reviewed techniques applied to direct toxins to cancer tissues.

Response: We have included a section providing a brief summary of the techniques discussed in the present updated review,  which indeed improves the Conclusions section.

In addition, a few references in the schemes have been corrected.